# An HSV-2 nucleoside-modified mRNA genital herpes vaccine containing glycoproteins gC, gD, and gE protects mice against HSV-1 genital lesions and latent infection

**Kevin P. Egan**[1], **Lauren M. Hook**[1], **Alexis Naughton**[1], **Norbert Pardi**[1], **Sita Awasthi**[1], **Gary H. Cohen**[2], **Drew Weissman**[1], **Harvey M. Friedman**[1] *

**1** Infectious Disease Division, Department of Medicine, Perelman School of Medicine, University of Pennsylvania, Philadelphia, Pennsylvania, United States of America, **2** Department of Basic and Translational Sciences, School of Dental Medicine, University of Pennsylvania, Philadelphia, Pennsylvania, United States of America

\* hfriedma@pennmedicine.upenn.edu

**Data Availability Statement:** The data are all available within the manuscript.

## Abstract

HSV-1 causes 50% of first-time genital herpes infections in resource-rich countries and affects 190 million people worldwide. A prophylactic herpes vaccine is needed to protect against genital infections by both HSV-1 and HSV-2. Previously our laboratory developed a trivalent vaccine that targets glycoproteins C, D, and E present on the HSV-2 virion. We reported that this vaccine protects animals from genital disease and recurrent virus shedding following lethal HSV-2 challenge. Importantly the vaccine also generates cross-reactive antibodies that neutralize HSV-1, suggesting it may provide protection against HSV-1 infection. Here we compared the efficacy of this vaccine delivered as protein or nucleoside-modified mRNA immunogens against vaginal HSV-1 infection in mice. Both the protein and mRNA vaccines protected mice from HSV-1 disease; however, the mRNA vaccine provided better protection as measured by lower vaginal virus titers post-infection. In a second experiment, we compared protection provided by the mRNA vaccine against intravaginal challenge with HSV-1 or HSV-2. Vaccinated mice were totally protected against death, genital disease and infection of dorsal root ganglia caused by both viruses, but somewhat better protected against vaginal titers after HSV-2 infection. Overall, in the two experiments, the mRNA vaccine prevented death and genital disease in 54/54 (100%) mice infected with HSV-1 and 20/20 (100%) with HSV-2, and prevented HSV DNA from reaching the dorsal root ganglia, the site of virus latency, in 29/30 (97%) mice infected with HSV-1 and 10/10 (100%) with HSV-2. We consider the HSV-2 trivalent mRNA vaccine to be a promising candidate for clinical trials for prevention of both HSV-1 and HSV-2 genital herpes.

**Funding:** HMF is the recipient of RO1 AI139618 from the National Institutes of Health. HMF, GHC, and DW are recipients of a grant from BioNTech SE. KPE is a post-doctoral trainee on T32 AI118684 (awarded to HMF). The Virus and Reservoirs Core of the Penn Center for AIDS Research provided support for qPCR studies through an NIH funded program (P30 AI045008). The funders had no role in study design, data collection and analysis, decision to publish, or preparation of the manuscript.

**Competing interests:** HMF, SA, GHC and DW are inventors on patents held by the University of Pennsylvania for protein (HMF, SA) and mRNA (HMF, SA, GHC, DW) vaccines for genital herpes. NP is also named on a patent describing use of nucleoside-modified mRNA-LNP as a vaccine platform. The authors have disclosed their interests fully to the University of Pennsylvania, and we have in place an approved plan for managing any potential conflicts arising from licensing of our patents.

## Author summary

Herpes simplex virus type 1 (HSV-1) is an important cause of genital herpes infection, although worldwide herpes simplex virus type 2 (HSV-2) is the most common cause. Herpes infections persist for life and there is no cure. A preventative vaccine is the best approach to reduce new genital herpes infections. An optimal vaccine should protect against both HSV-1 and HSV-2 infection. Our vaccine targets HSV-2 glycoproteins C, D, and E administered either as proteins or mRNA encapsulated in lipid nanoparticles. We compared the vaccine delivered as mRNA or proteins for prevention of HSV-1 genital infection in mice. Both vaccines prevented genital disease but the mRNA vaccine was better at limiting virus replication in the genital tract. We then compared protection by the HSV-2 mRNA vaccine against genital HSV-1 and HSV-2 infection. Mice infected with either virus were totally protected from genital disease. Importantly, in two experiments, the mRNA vaccine prevented HSV invasion of the dorsal root ganglia, the site of virus latency, in 39/40 (97.5%) mice infected with either HSV-1 or HSV-2. We conclude that the HSV-2 trivalent mRNA vaccine provides potent protection against both HSV-1 and HSV-2 genital infection and is a promising vaccine candidate for human trials.

## Introduction

Herpes simplex virus type 2 (HSV-2) genital infection affects a half-billion people worldwide [1]. Herpes simplex virus type 1 (HSV-1) genital infection affects an additional 190 million people [2, 3]. Fifty percent of first-time genital herpes infections in resource-rich countries are caused by HSV-1. Successful public health measures have reduced acquisition of oral HSV-1 at a young age, which leaves many people susceptible to genital HSV-1 in these regions [2]. First-episode HSV-2 genital herpes rates are declining in resource-rich countries while first-episode HSV-1 infection rates have remained stable for 25 years [4]. Although HSV-1 is an important cause of first-time genital infections, HSV-1 is much less likely to cause recurrent genital infections than HSV-2 [5]. As a result, HSV-2 accounts for 95% of genital herpes disease worldwide [3]. This epidemiology suggests that HSV-2 is the dominant genital pathogen but ideally a prophylactic genital herpes vaccine will protect against both HSV-1 and HSV-2.

No prophylactic genital herpes vaccine is currently approved despite major efforts involving three large phase 3 trials [6–8]. However, multiple observations support continued efforts to develop an effective vaccine. First, reinfection with a different HSV-2 isolate is uncommon [9]. Second, prior oral infection with HSV-1 partially reduces the severity of a first-time HSV-2 genital infection, while prior HSV-2 genital infection provides perhaps complete protection against first-time genital HSV-1 infection [10]. Third, mothers with recurrent genital herpes infection have a reduced risk of transmitting the virus to their newborns during labor and delivery compared to mothers with first-time genital herpes infection [11]. This reduced risk is attributed to antibodies transferred from mother to infant transplacentally, a concept that is supported by vaccine studies in mice that demonstrate maternal antibodies protect newborns [12–15]. Fourth, results from the Herpevac Trial for Women suggest that an effective vaccine for genital herpes is possible based on protection provided by HSV-2 glycoprotein D (gD2) vaccine against genital HSV-1 infection. The gD2 vaccine was 77% efficacious against HSV-1 genital disease after 3 immunizations, although not efficacious against HSV-2 [8]. Vaccine-induced antibodies correlated with protection against HSV-1 infection [16]. A subset of subjects were evaluated for neutralizing antibody titers that were higher to HSV-1 than HSV-2,

which may explain the better protection against HSV-1 [17]. Another possible explanation for better protection against HSV-1 is that genital HSV-1 may be less virulent than HSV-2 [18].

HSV immune evasion represents a barrier to a successful vaccine. HSV-1 and HSV-2 glycoprotein C (gC1, gC2) bind complement C3b and limit its ability to participate in activating the complement cascade, an important innate immune defense [19–22]. HSV-1 and HSV-2 glycoprotein E (gE1, gE2) function as immunoglobulin G (IgG) Fc receptors that bind the IgG Fc domain of an antibody that is also bound by its F(ab')$_2$ domain to an HSV antigen [23, 24]. These immune evasion molecules prevent antibodies and complement from participating in host defense by inhibiting complement activation and antibody-dependent cellular cytotoxicity. Our vaccine for preventing HSV-2 genital infection consists of three HSV-2 glycoproteins, gC2, gD2, and gE2 that are expressed on the virion envelope and at the surface of infected cells and thus potentially accessible to antibodies that block their functions. Antibodies to gD2 block virus entry, antibodies to gC2 neutralize virus and block C3b binding, while antibodies to gE2 block cell-to-cell spread and immune evasion [25–27].

We compared two vaccine platforms for delivering the HSV-2 trivalent gC2/gD2/gE2 vaccine. One platform expresses the antigens in baculovirus and administers purified proteins with CpG oligonucleotides and aluminum hydroxide (alum) as adjuvants [25]. The other platform expresses the same immunogens as nucleoside-modified mRNA in lipid nanoparticles (LNP) [26]. The mRNA molecule is modified to improve stability and prevent innate immune sensors from inhibiting translation [28]. Nucleoside-modified mRNA-LNP stimulates potent T follicular helper cell and germinal B cell responses that result in high titer and durable antibody responses [29]. The HSV-2 trivalent nucleoside-modified mRNA-LNP (mRNA) vaccine outperformed the trivalent protein-CpG/alum (protein) vaccine in producing higher titers of neutralizing antibodies and antibodies that block gC2 and gE2 immune evasion activities, and in protecting mice and guinea pigs against intravaginal HSV-2 infection [26].

A previous report demonstrated that gD2 subunit protein administered with monophosphoryl lipid A and alum as adjuvants provided better protection against HSV-1 genital infection than HSV-2 in female cotton rats [18]. The authors attributed better protection against HSV-1 in part to the lower virulence of HSV-1 than HSV-2 in this model. Another study that used gD2 subunit protein with monophosphoryl lipid A and alum reported slightly better protection against genital HSV-1 infection than HSV-2 in guinea pigs [30]. The cotton rat and guinea pig results are consistent with the outcome reported in the Herpevac Trial for Women [8]. We previously reported that immunization with the HSV-2 trivalent protein vaccine protects against HSV-1 genital lesions and recurrent HSV-1 genital shedding after intravaginal infection of guinea pigs [31]. Since performing those guinea pigs studies, we determined that the HSV-2 mRNA vaccine version outperforms the protein vaccine in preventing HSV-2 genital infection [26]. The goals of the current study were: 1) to determine whether the HSV-2 mRNA vaccine outperforms the protein vaccine in protecting against HSV-1 genital infection; and 2) to evaluate whether the HSV-2 mRNA vaccine protects as well against HSV-1 as against HSV-2 genital infection.

## Results

### Intravaginal HSV-1 infection in naïve mice

We evaluated the susceptibility of naïve, unimmunized mice to HSV-1 intravaginal infection to determine the optimal challenge dose to be used in subsequent immunization experiments. Eight- to nine-week-old naïve female BALB/c mice were infected with varying concentrations of HSV-1 strain NS. Mice were assessed for clinical disease and euthanized when humane endpoints were reached. The HSV-1 lethal dose for 50% of the animals (LD$_{50}$) was $3.73 \times 10^4$

plaque-forming units (PFU) (Fig 1A). We selected an HSV-1 infection dose of $2\times10^6$ PFU (54 $LD_{50}$) as a potent challenge to assess vaccine efficacy.

## Intravaginal HSV-1 challenge of immunized mice

We compared vaccine efficacy of the HSV-2 gC2/gD2/gE2 nucleoside-modified mRNA-LNP (mRNA) formulation with the baculovirus gC2/gD2/gE2 protein CpG/alum (protein) vaccine. The two vaccines express the identical gC2, gD2, and gE2 amino acids. Female eight- to nine-week old BALB/c mice were immunized twice at 28-day intervals with Poly(C) RNA-LNP (Poly(C) control group) or with mRNA, while animals that received the protein vaccine were

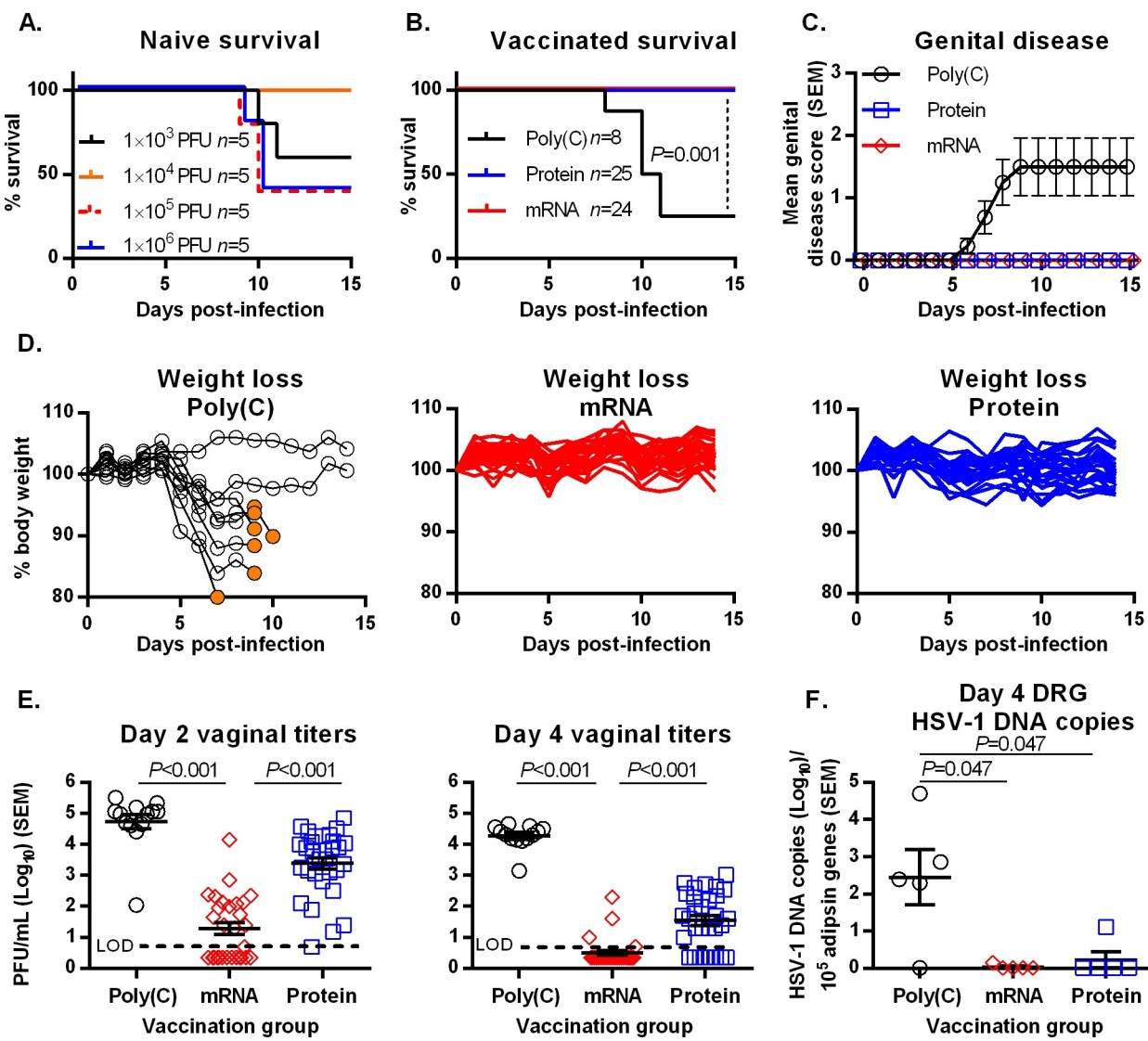

**Fig 1. HSV-2 gC2/gD2/gE2 trivalent immunization protects mice from intravaginal HSV-1 disease.** (A) Survival curve of naïve female BALB/c mice infected with varying concentrations of HSV-1. (B-F) BALB/c mice were immunized with Poly(C), trivalent mRNA, or trivalent protein and challenged intravaginally with HSV-1 at $2\times10^6$ PFU. (B) Survival curves. *P* values were calculated by the log-rank test. (C) Mean genital disease scores in the Poly(C), protein and mRNA groups. (D) Weight loss in individual mice in the Poly(C), protein, and mRNA groups. Orange filled circles indicate mice that were sacrificed after reaching humane endpoints. (E) Vaginal swab titers on day two (left) and day four (right) post-infection. Dashed line indicates the assay limit of detection (LOD) of 5 PFU/mL. *P* values compare the mean virus titers. (F) HSV-1 DNA copy number in DRG four days post-infection. *P* values in E and F were calculated by the two-tailed Mann-Whitney test.

immunized three times at two-week intervals. HSV-1 ($2\times10^6$ PFU) was inoculated intravaginally one month after the final immunization. Mice were observed for survival and clinical disease as measured by genital lesions and weight loss. In the Poly(C) RNA-LNP control group, 6/8 (75%) mice died between days 7–11, while no mouse in the protein (0/25) or mRNA (0/24) group died (Fig 1B). 6/8 (75%) mice in the Poly(C) group developed genital disease and lost weight, while no mouse in the mRNA or protein group developed genital disease or lost weight (Fig 1C and 1D).

Mice were evaluated for subclinical infection by obtaining vaginal swabs for virus cultures, measuring HSV-1 DNA copy number in dorsal root ganglia (DRG), and evaluating genital tract tissues for histopathology and immunohistochemistry. On day two post-infection, virus was isolated from 13/13 (100%) mice in the Poly(C) group (mean $\log_{10}$ titer 4.73 PFU/mL), 16/29 (55%) mice in the mRNA group (mean $\log_{10}$ titer 1.29 PFU/mL), and 30/30 (100%) mice in the protein group (mean $\log_{10}$ titer 3.38 PFU/mL) (Fig 1E, left). By day four, the number of animals with positive vaginal titers declined in the immunized animals but not in the Poly(C) controls. Virus was isolated from 13/13 (100%) Poly(C)-immunized mice (mean $\log_{10}$ titer 4.27 PFU/mL), 4/29 (14%) in the mRNA vaccinated group (mean $\log_{10}$ titer 0.49 PFU/mL), and 22/30 (73%) in the protein group (mean $\log_{10}$ titer 1.54 PFU/mL) (Fig 1E, right). Five mice from each group were euthanized four days post-challenge and HSV-1 DNA copy number in lumbosacral DRG was measured by qPCR. HSV-1 genomes were detected in the DRG from 4/5 (80%) mice in the Poly(C) group (mean $\log_{10}$ 2.45 DNA copies) compared to 1/5 (20%) in the mRNA group (mean $\log_{10}$ 0.038 DNA copies) and 1/5 (20%) in the protein group (mean $\log_{10}$ 0.23 DNA copies) (Fig 1F). These results support vaccine efficacy for both mRNA and protein formulations, but the mRNA vaccine was more potent based on fewer mice with positive virus titers on days two and four post-infection and lower mean virus titers in the mRNA group.

## Histopathology and immunohistochemistry after HSV-1 intravaginal challenge

As another approach to compare protection provided by the mRNA and protein vaccines, we performed histopathology and immunohistochemistry for HSV-1 antigens on genital tract tissues harvested four days post-infection. The normal histology of the female genital tract in an uninfected, non-immunized mouse is shown in Fig 2A (Naïve). 5/5 (100%) Poly(C) immunized animals (controls) infected with HSV-1 at $2\times10^6$ PFU developed large ulcerations (white arrowheads), necrosis and inflammatory debris in the vaginal epithelial lining with abundant inflammatory infiltrates in the lamina propria (Fig 2A, Poly(C)). Characteristic viral inclusion bodies including multinucleated cells, HSV Cowdry type A viral inclusions, and nuclei with chromatin margination were present (Fig 2B, Poly(C)). 3/5 (60%) mice immunized with protein had some histopathologic evidence of infection denoted by areas of thickened vaginal epithelium (white brackets in Fig 2A, protein) and superficial erosions without ulcerations (white arrows in Fig 2A, protein). In mRNA-immunized mice, the genital tract tissues were nearly devoid of pathology. Normal tissues were detected in 4/5 (80%) mice (Fig 2A, mRNA). One animal had a single focus of infection that looked similar to the area shown for the protein group (Fig 2A, protein).

We performed immunohistochemistry to detect HSV-1 antigens in genital tract tissues. Foci of infection were counted along the entire length of the vagina at two depths separated by 100 μm. Uninfected naïve mice were negative for HSV-1 antigens (Fig 2C, naïve). 5/5 (100%) mice in the Poly(C) group had multiple large foci of HSV-1 antigen that coincided with areas of infection detected by histopathology (Fig 2C, Poly(C)). 3/5 (60%) mice in the protein group had one or two foci that were positive for HSV-1 antigen, although the infection was greatly

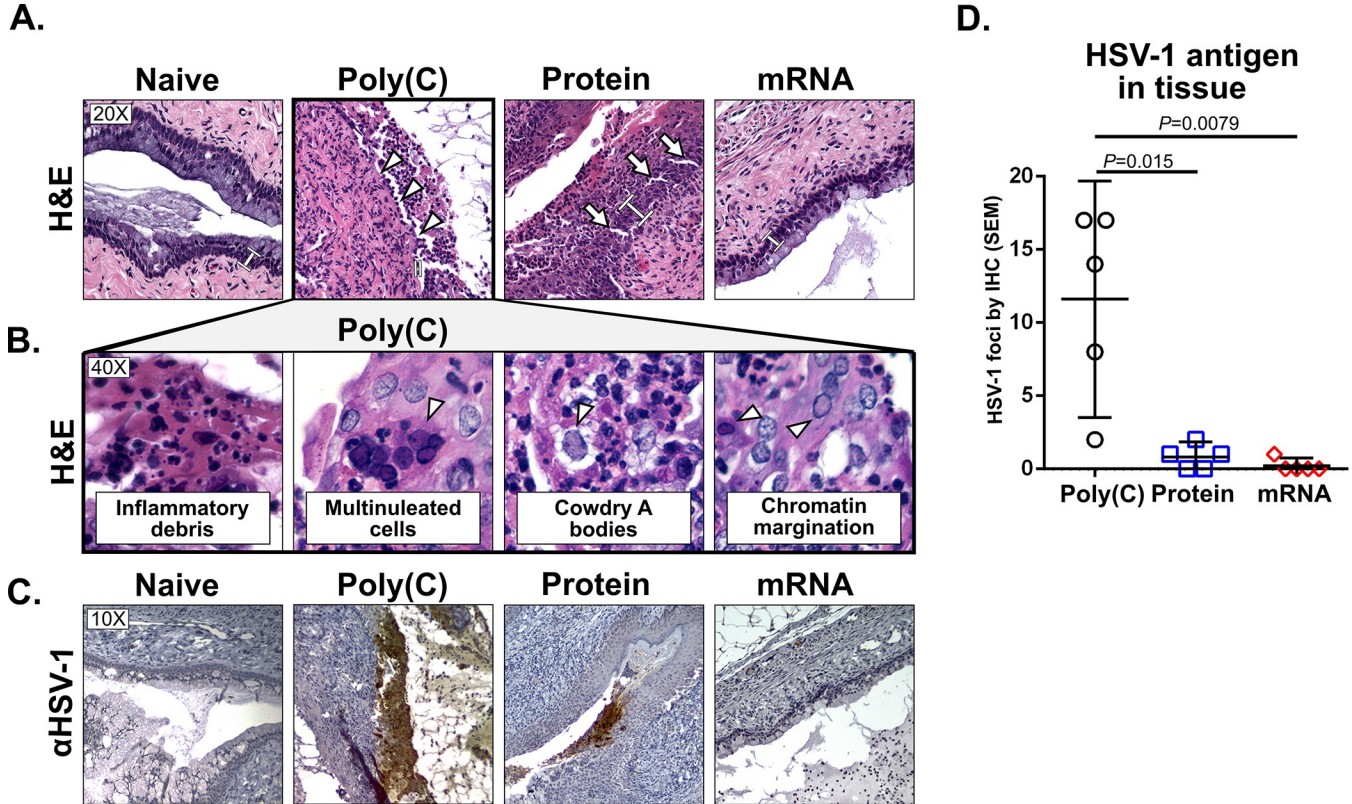

**Fig 2. Histopathology and immunohistochemistry four days after HSV-1 intravaginal infection.** (A) Hematoxylin and Eosin staining of genital tract tissues from naïve mice that were not immunized or infected, or from mice that were immunized with Poly(C) RNA (control), protein, or mRNA and infected with HSV-1 at $2\times10^6$ PFU. White arrowheads indicate ulcerations in the epithelial cell layer. White arrows indicate erosion in the epithelial cell layer. White brackets indicate epithelial cell layer thickness. (B) Inflammatory debris, multinucleated cells (white arrowhead), inclusion bodies (white arrowhead) and chromatin margination (white arrowhead) in Poly(C) group. Images were taken with 20X and 40X objectives. (C) Immunohistochemistry for HSV-1 antigen using a 10X objective. (D) The number of foci positive for HSV-1 antigen in the Poly(C), protein and mRNA groups. *P* values were obtained using the two-tailed Mann-Whitney test. Sample size is *n* = 5 mice per group.

reduced compared to Poly(C) (Fig 2C, protein). In the mRNA group, a single focus of HSV-1 antigen was noted in one mouse that corresponded to the area detected by histopathology. No HSV-1 antigen was identified in the remaining four mice in the mRNA group (Fig 2C, mRNA). Images of HSV-1 antigen-positive areas were captured using a 2X objective and the number of foci counted. A significant reduction was noted in the number of HSV-1 foci in the protein and mRNA vaccinated groups compared to the Poly(C) controls (Fig 2D). We conclude that high dose HSV-1 infection produced extensive genital tract disease in Poly(C)-immunized mice, while the protein and mRNA vaccines provided potent protection. The mRNA vaccine outperformed the protein formulation based on fewer foci of infection detected in genital tract tissues (Fig 2D), and lower day two and day four vaginal virus titers (Fig 1E); however, protection by the mRNA vaccine was not complete based on a single focus of genital infection detected by histopathology and immunohistochemistry in 1/5 mice.

## Comparing protection provided by HSV-2 trivalent nucleoside-modified mRNA-LNP against HSV-1 and HSV-2 intravaginal challenge

We previously reported that 64/64 mice immunized with the mRNA vaccine and challenged with HSV-2 at $5\times10^3$ PFU (~ 275 LD$_{50}$) or $5\times10^4$ PFU (~ 2,750 LD$_{50}$) had negative day two

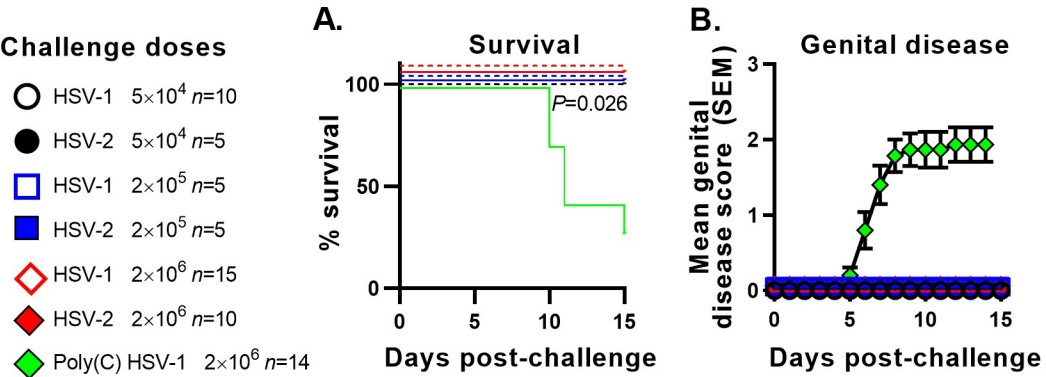

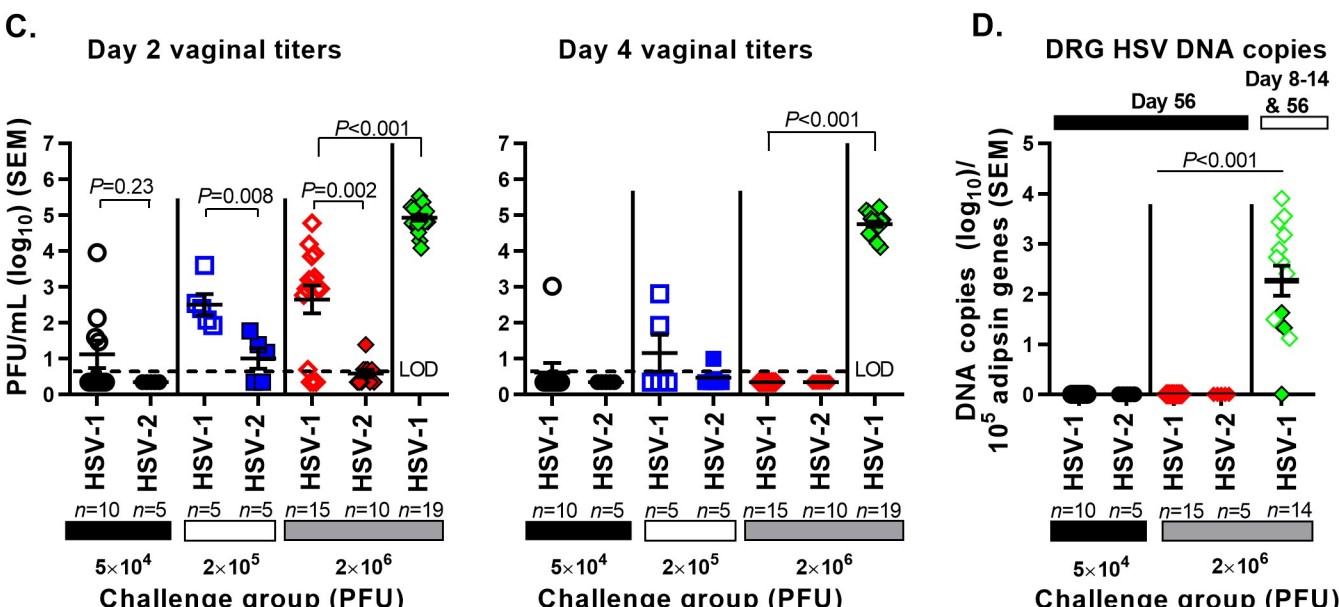

**Fig 3. Protection provided by HSV-2 trivalent nucleoside-modified mRNA-LNP against HSV-1 and HSV-2 intravaginal challenge.** Mice were immunized with Poly(C) (control) or HSV-2 mRNA and infected with matched doses of HSV-1 or HSV-2. (A) Survival curves. *P* value was calculated using the log-rank test. (B) Mean genital disease score after challenge. (C) Day two vaginal titers and day four vaginal titers. Dotted lines indicate the assay limit of detection (LOD) of 5 PFU/mL. (D) HSV DNA copy number in DRG of mice sacrificed at humane endpoints or at the end of the study. Green diamonds represent Poly(C) vaccinated mice sacrificed at humane endpoints (open symbols, 10 animals) or day 56 at the end of experiment (closed symbols, 4 animals with two of the closed symbols superimposed at 1.3 $\log_{10}$). *P* values in (C-D) were calculated using the two-tailed Mann-Whitney test.

and day four cultures [26]. In Fig 1, we reported that some mice immunized with the mRNA vaccine had positive day two cultures (16/29 mice) and day four cultures (4/29 mice) when infected with HSV-1 at $2\times10^6$ PFU. The challenge dose for HSV-1 was 40-fold higher than the highest challenge dose for HSV-2, but 50-fold lower in terms of $LD_{50}$. Therefore, we next evaluated protection provided by the mRNA vaccine when immunized mice were challenged with comparable HSV-1 and HSV-2 doses.

Immunized mice were challenged intravaginally with HSV-1 or HSV-2 at three doses, $5\times10^4$, $2\times10^5$, or $2\times10^6$ PFU. As a control, mice were immunized with Poly(C) and challenged with $2\times10^6$ PFU HSV-1. The mRNA-immunized mice were totally protected against death and genital lesions at each HSV-1 and HSV-2 challenge dose, while 10/14 (71%) mice in the

Poly(C) group developed genital disease and died (Fig 3A and 3B). At the lowest challenge dose of $5\times10^4$ PFU, vaginal swabs were positive on day two in 4/10 (40%) mice challenged with HSV-1 (mean $\log_{10}$ titer 1.26 PFU/mL), while 0/5 mice challenged with HSV-2 were positive (Fig 3C, left), consistent with our prior report of total protection in mRNA-immunized mice challenged with $5\times10^4$ HSV-2 [26]. At the next higher dose of $2\times10^5$ PFU, 5/5 (100%) mice challenged with HSV-1 had positive titers on day two compared to 3/5 (60%) mice challenged with HSV-2, and significantly higher titers of virus were present in the HSV-1 challenged mice (mean $\log_{10}$ titer 2.51 PFU/mL) than in the HSV-2 group (mean $\log_{10}$ titer 1.01 PFU/mL) (Fig 3C, left). At the highest challenge dose of $2\times10^6$ PFU, 12/15 (80%) mice in the HSV-1 group had positive vaginal titers compared to 5/10 (50%) mice in the HSV-2 group. Once again, the mean titers in the HSV-1 group were significantly higher compared to HSV-2 (mean $\log_{10}$ titers 2.65 compared to 0.59 PFU/mL) (Fig 3C, left). HSV-1 day two titers were higher in this experiment than in Fig 1E; however, in both experiments, infection with $2\times10^6$ PFU produced higher day two titers for HSV-1 than HSV-2 (Fig 3C, left). The day two vaginal titers in the Poly(C) group infected with HSV-1 at $2\times10^6$ PFU were significantly higher than day two titers in mRNA-immunized mice challenged with HSV-1 at this dose (Fig 3C, left).

HSV-1 and HSV-2 titers in mRNA-immunized mice were considerably lower on day four post-infection. Only 3/30 (10%) mice were positive in the combined HSV-1 groups and 1/20 (5%) positive in the combined HSV-2 groups (Fig 3C, right). The HSV-1 day four vaginal titers were significantly higher in the Poly(C) control group than in the mRNA group challenged at $2\times10^6$ PFU (Fig 3C, right). These results indicate that higher day two vaginal titers in the HSV-1 group was the only significant difference between protection provided against HSV-1 and HSV-2.

DRG were harvested at the time of humane euthanasia between days 8–14 for the 10/14 mice that succumbed to HSV-1 infection in the Poly(C) group or on day 56 for the four surviving Poly(C) animals and all mice in the mRNA groups challenged with HSV-1 or HSV-2 at $5\times10^4$ or $2\times10^6$ PFU. No HSV DNA was detected in the DRG of any mRNA-immunized mouse, while 13/14 (93%) mice in the Poly(C) group were positive for HSV-1 DNA. The HSV-1 DNA copy number in the four surviving animals in the Poly(C) group is denoted by filled diamonds, while those euthanized earlier for humane reasons are indicated by open diamonds (Fig 3D). Therefore, in this side-by-side comparison, the HSV-2 mRNA vaccine completely protected mice from death, genital disease and DRG infection after challenge with HSV-1 or HSV-2 at doses between $5\times10^4$ and $2\times10^6$ PFU, but the mRNA vaccine provided better protection against day two vaginal virus replication after HSV-2 challenge than HSV-1.

## Intravaginal HSV-2 in naïve mice

To further assess differences in protection against HSV-1 and HSV-2, we evaluated HSV-2 infection in naïve, unimmunized mice. Our prior studies demonstrated that HSV-2 was lethal in mock-immunized mice at titers much lower than reported here for HSV-1 (Fig 1A) [26]. To confirm the HSV-2 $LD_{50}$, we infected naïve mice using serial 10-fold concentrations ranging from $1.5\times10^1$ to $1.5\times10^4$ PFU HSV-2 (Fig 4A). The HSV-2 $LD_{50}$ was 19 PFU, consistent with our prior study, and approximately 2,000-fold lower than the HSV-1 $LD_{50}$ ($3.73\times10^4$ PFU, Fig 1A). We next compared genital tract replication of the two viruses in naïve mice that were infected with comparable doses of HSV-1 and HSV-2. Mice infected with HSV-1 at $1\times10^3$ or $1\times10^4$ PFU had similar day 2 vaginal titers as mice infected with HSV-2 at $1.5\times10^3$ or $1.5\times10^4$ PFU (Fig 4B). Therefore, virus replication in the genital tract of naïve mice using similar, although slightly higher doses for HSV-2, does not explain the lower HSV-2 titers on day two in immunized mice.

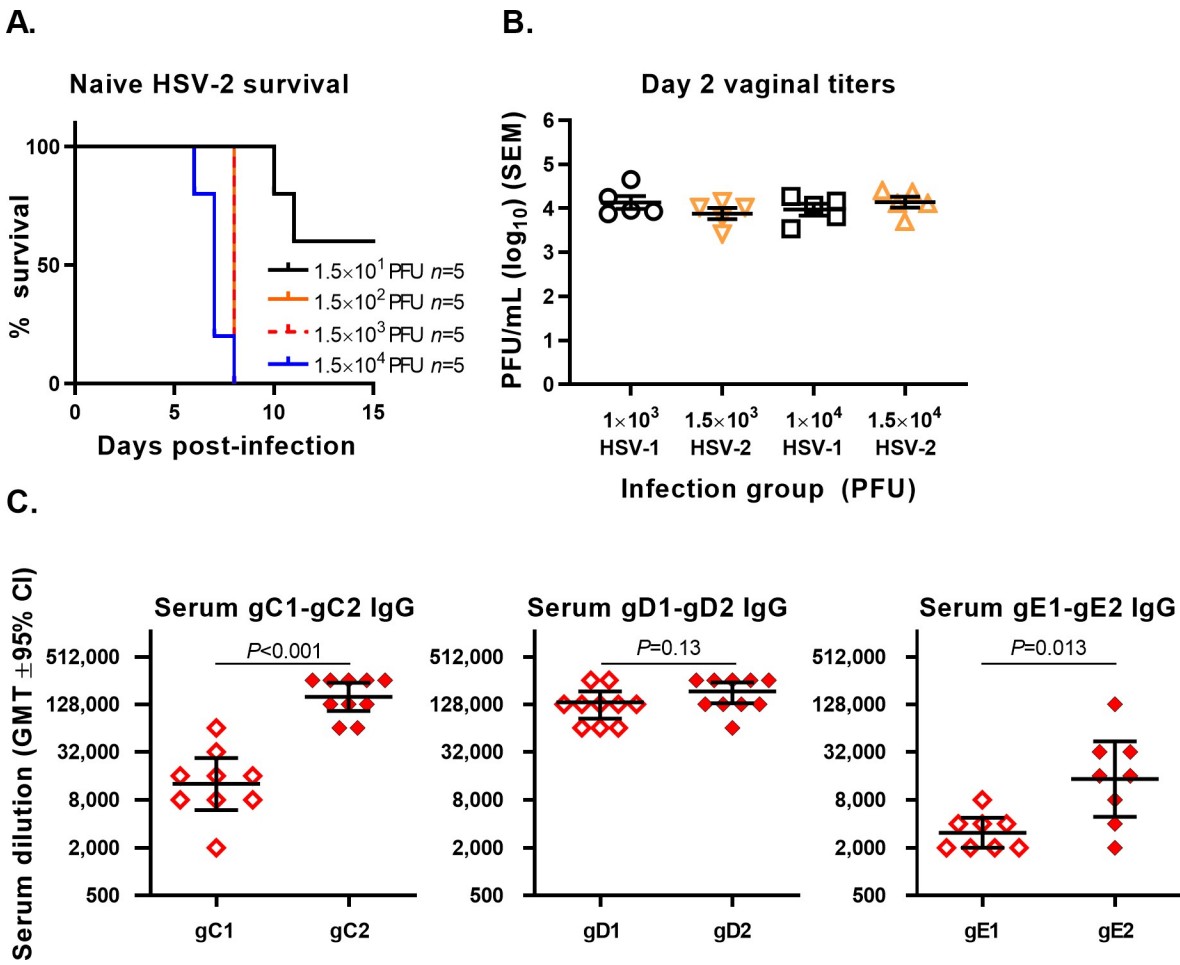

**Fig 4. Intravaginal infection of naïve, unimmunized mice.** (A) Survival curves after HSV-2 infection in naïve mice to determine $LD_{50}$. (B) Day two vaginal virus titers after infection with HSV-1 at $1 \times 10^3$ or $1 \times 10^4$ PFU or HSV-2 at $1.5 \times 10^3$ or $1.5 \times 10^4$ PFU. (C) Sera from mRNA vaccinated animals evaluated for IgG ELISA titers to gC1 or gC2, gD1 or gD2, and gE1 or gE2. *P* values were calculated by two-tailed Student's t test.

We evaluated whether type-specific immunity may explain the better protection provided by the HSV-2 mRNA vaccine against HSV-2 than HSV-1. We performed IgG ELISA assays to assess cross-immunogenicity. The HSV-2 gC2, gD2, gE2 amino acid sequences share 65% identity for gC1, 82% for gD1 and 73% for gE1, respectively. Sera were obtained from HSV-2 mRNA-immunized mice after the final immunization, prior to virus infection. ELISA IgG endpoint titers were significantly higher to gC2 than gC1 and to gE2 than gE1, while gD2 and gD1 titers did not differ significantly (Fig 4C). The type-specific immunity against gC2 and gE2 may explain the better protection provided by the HSV-2 mRNA vaccine against HSV-2 than HSV-1 in this study.

## Summary

Our goals were to compare mRNA with protein immunization for protection against HSV-1 genital challenge and to determine whether an HSV-2 vaccine was more efficacious against HSV-1 or HSV-2 genital challenge. Table 1 summarizes the results presented in Figs 1 and 2

**Table 1. HSV-2 trivalent mRNA or protein vaccine protect against HSV-1 at $2\times10^6$ PFU[*].**

| Outcome | Poly(C) | | mRNA | | Protein |
|---|---|---|---|---|---|
| Clinical disease | | | | | |
| Death | 16/22 (73%) | $P<0.0001$ | 0/24 (0%) | | 0/25 (0%) |
| Genital disease | 16/22 (73%) | $P<0.0001$ | 0/24 (0%) | | 0/25 (0%) |
| Subclinical infection | | | | | |
| Day 2 titers | 27/27 (100%) | $P<0.0001$ | 16/29 (55%) | $P<0.0001$ | 30/30 (100%) |
| Day 4 titers | 27/27 (100%) | $P<0.0001$ | 4/29 (14%) | $P<0.0001$ | 22/30 (73%) |
| Day 4 histopathology & immunohistochemistry | 5/5 (100%) | $P = 0.0476$ | 1/5 (20%) | | 1/5 (20%) |
| HSV-1 DNA in DRG[#] | 17/19 (89%) | $P = 0.0065$ | 1/5 (20%) | | 1/5 (20%) |
| Totals | | | | | |
| Clinical & subclinical | 27/27 (100%) | $P<0.0001$ | 16/29 (55%) | $P<0.0001$ | 30/30 (100%) |
| Clinical & only DRG as subclinical | 23/27 (85%) | $P<0.0001$ | 1/29 (3%) | | 1/30 (3%) |

[*] Poly(C) animals are from Figs 1–3; mRNA animals are from Figs 1 and 2.

[#] DRG were harvested on day 4, at time of euthanasia or at the end of experiment. *P* values ≤0.05 are indicated. *P* values were calculated using two-tailed Fisher's exact test.

that compared mRNA and protein vaccines in mice challenged with HSV-1 at $2\times10^6$ PFU. The only significant difference was better protection by the mRNA vaccine against day two and day four vaginal titers. Table 2 combines results presented in Fig 3 from mRNA-immunized animals infected with HSV-1 or HSV-2 at three doses. Significant differences were noted based on fewer animal in the HSV-2 group with positive day two vaginal titers. The clinical relevance of detecting positive vaginal titers on day two or day four is unknown, while HSV DNA in DRG in mice is an important marker for latent infection and risk for recurrent infection. Therefore, outcomes that are clearly clinically meaningful include death, genital disease and HSV DNA in DRG. Based on results shown in Tables 1 and 2 in mRNA-immunized mice, the mRNA vaccine prevented death and genital disease in 54/54 (100%) mice infected with HSV-1 and 20/20 (100%) with HSV-2, and prevented infection of the dorsal root ganglia in 29/30 (97%) mice infected with HSV-1 and 10/10 (100%) with HSV-2. Therefore, the mRNA vaccine was highly protective against both HSV-1 and HSV-2 for outcomes that are clinically meaningful.

**Table 2. HSV-2 trivalent mRNA protection against HSV-1 or HSV-2 [*].**

| Outcome | HSV-1 | | HSV-2 |
|---|---|---|---|
| Clinical disease | | | |
| Death | 0/30 (0%) | | 0/20 (0%) |
| Genital lesions | 0/30 (0%) | | 0/20 (0%) |
| Subclinical infection | | | |
| Day 2 titers | 21/30 (70%) | $P = 0.045$ | 8/20 (40%) |
| Day 4 titers | 3/30 (10%) | | 1/20 (5%) |
| HSV DNA in DRG day 56 | 0/25 (0%) | | 0/10 (0%) |
| Totals | | | |
| Clinical & any subclinical | 21/30 (70%) | $P = 0.045$ | 8/20 (40%) |
| Clinical & only DRG as subclinical | 0/30 (0%) | | 0/20 (0%) |

[*] Animals are from Fig 3. Values represent combining all animals challenged at $5\times10^4$, $2\times10^5$, and $2\times10^6$ PFU. *P* values ≤0.05 are indicated. *P* values were calculated using two-tailed Fisher's exact test.

## Discussion

An effective prophylactic vaccine is the best approach to limit new HSV-1 and HSV-2 genital infections. We previously reported that an HSV-2 gC2, gD2, gE2 nucleoside-modified mRNA-LNP vaccine protects mice and guinea pigs from HSV-2 intravaginal infection and that the mRNA vaccine outperformed the same antigens expressed using baculovirus proteins administered with CpG and alum [26]. Multiple immune responses were superior in the trivalent mRNA compared with the trivalent protein group, including serum and vaginal IgG ELISA titers, HSV-2 neutralizing antibody titers, antibodies that block gC2 and gE2 immune evasion domains, antibody responses to gD2 epitopes involved in virus entry and cell-to-cell spread, CD4$^+$ T follicular helper and germinal center B cell responses [26]. The induction of T follicular helper cells occurs independent of Toll-like receptors and type 1 interferon [29]. The T follicular helper cells stimulate germinal center B cell responses that likely account for the high titers of neutralizing antibodies and antibodies that block C3b binding and IgG Fc binding that are the proposed mechanisms for the improved protection with nucleoside-modified mRNA-LNPs compared to adjuvanted protein.

Here, we evaluated whether the mRNA vaccine protected mice against intravaginal HSV-1 challenge and compared protection against HSV-1 and HSV-2. Both mRNA and protein formulations completely prevented death, weight loss and genital disease after HSV-1 infection. Both formulations reduced the number of mice positive for HSV-1 DNA in DRG and significantly reduced HSV-1 DNA copies present. However, mice immunized with the mRNA vaccine had lower day two and day four vaginal titers and fewer animals had evidence of HSV-1 infection in the female genital tract by histopathology and immunohistochemistry. Comparing mRNA immunization for protection against HSV-1 or HSV-2, the vaccine was more protective against HSV-2 based on day two vaginal titers. We previously reported total protection against $5\times10^3$ or $5\times10^4$ HSV-2 challenge in mRNA-immunized mice [26]. Here, we detected positive vaginal titers on day two and day four after HSV-2 challenge at $2\times10^5$ and $2\times10^6$ PFU. Despite these positive titers, the trivalent mRNA vaccine provided complete protection against HSV-2 clinical disease and no viral genomes were detected in lumbosacral DRG at these challenge doses that were 40-fold higher than previously tested (100,000 LD$_{50}$) [26]. The absence of HSV-2 DNA in DRG suggests that vaccine-induced immunity may prevent DRG infection despite local virus replication in genital tract tissues, or possibly that the qPCR assay may not be sufficiently sensitive to detect low level DRG infection despite being able to detect one copy of HSV-2 DNA per $10^5$ copies of adipsin DNA.

We proposed that pre-clinical genital herpes vaccines should meet a high standard of protection in animal models before proceeding to human clinical trials [32]. Our results support better protection by the HSV-2 mRNA vaccine against HSV-2 than HSV-1, raising a question whether the protection against HSV-1 is good enough to proceed to human trials. We propose that the answer is affirmative based on the following reasoning. First, no animal (0/54) that was immunized with the mRNA vaccine and challenged with HSV-1 developed genital lesions, weight loss or died. Second, only 1/30 (3%) mice had HSV-1 DNA detected in DRG when challenged. Third, the evidence of breakthrough infections is based on day two and day four vaginal titers. The histopathology and immunohistochemistry identified a focus of infection in 1/5 mRNA-immunized mice infected at $2\times10^6$ HSV-1, indicating that day two titers are not merely residual input virus. However, DRG infection was rare, suggesting potent protection when using DRG as a marker for vaccine efficacy. Fourth, HSV-1 comprises only 5% of the global burden of genital herpes [3]. HSV-1 genital infections reactivate at a lower frequency than HSV-2 [5]. Individuals with dual oral and genital HSV-1 infections have many fewer episodes of genital reactivation than oral [33]. Taken together, we consider the protection

provided by the HSV-2 mRNA vaccine against HSV-1 meets our standards for proceeding to human trials.

The Herpevac Trial for Women reported that the HSV-2 gD2 vaccine provided better protection against HSV-1 than HSV-2 [8]. Two pre-clinical studies in cotton rats and guinea pigs also reported somewhat better protection by gD2 against HSV-1 than HSV-2 [18, 30]. In contrast, we noted that the HSV-2 trivalent mRNA vaccine protects better against HSV-2 than HSV-1. We offer the following explanations for these differences. First, HSV-1 is less virulent than HSV-2 in the female genital tract of mice and possibly of women. At comparable challenge doses, HSV-1 is less likely to produce genital disease than HSV-2; therefore, it is not surprising that a gD2 vaccine with 82% amino acid identity with gD1 protected better against HSV-1 genital disease than HSV-2. Second, our vaccine candidate contains gC2 and gE2 immunogens in addition to gD2. Both gC2 and gE2 are relatively type-specific antigens, which may explain the enhanced protection by the trivalent vaccine against HSV-2. Despite adding two rather type-specific antigens, protection was only slightly better against HSV-2 than HSV-1, supporting the concept that it is easier to protect against genital HSV-1 than HSV-2. Our efforts with mRNA to date have focused on preventing genital herpes; however, we are also interested in evaluating whether mRNA immunization will be efficacious if used as a therapeutic vaccine [26]. We conclude that the HSV-2 trivalent mRNA vaccine is a promising candidate for clinical trials for preventing both HSV-1 and HSV-2 genital herpes.

## Materials and Methods

### Ethics statement

All animal studies were conducted under protocol 805187 approved by the University of Pennsylvania Institutional Animal Care and Use Committee. The authors strictly followed the "Guide for the Care and Use of Laboratory Animals" by the Committee on Care of Laboratory Animal Resources Commission on Life Sciences, National Research Council. The animal facilities are fully accredited and certified by the American Association for Accreditation of Laboratory Animal Care (AAALAC).

### Production of bac-gC2, bac-gD2 and bac-gE2 subunit proteins in Sf9 cells

Bac-gC2(426t) expresses gC2 amino acids 27–426, bac-gD2(306t) expresses gD2 amino acids 26–333, and bac-gE2(24-405t) expresses gE2 amino acids 24–405 [27, 34, 35]. Each protein subunit extends from the first amino acid after the signal sequence to just prior to the transmembrane domain.

### Production of mRNA and formulation in lipid nanoparticles

Trivalent nucleoside modified mRNAs encoding the same amino acid sequence as the baculovirus-produced proteins were synthesized as previously described [26]. Poly(C) RNA was used as a control and is similar to antigen-encoding mRNA in that it does not induce type 1 interferons or proinflammatory cytokines and is immunologically silent [29].

### Immunizations

Female BALB/c mice (Charles River Laboratories) were 8–9 weeks old when first immunized. The gC2, gD2, and gE2 protein antigens were administered using 5 μg of each protein per mouse. The proteins were individually incubated at room temperature for two hours with 16.7 μg of CpG oligonucleotide 5'-TCCATGACGTTCCTGACGTT-3' (Coley Pharmaceutical) and 25 μg of alum per μg protein (Alhydrogel; Accurate Chemical and Scientific Corp.). The

proteins were combined just prior to immunization in a total volume of 50 μl. Mice were immunized three times at two-week intervals into the gastrocnemius muscle [27]. For the mRNA immunizations, 10 μg of each gC2, gD2, and gE2 mRNA was combined prior to LNP encapsulation. Mice were immunized two times at four-week intervals into the gastrocnemius muscle in a volume of 50 μl. Thirty days after the final immunization mice were bled and infected intravaginally.

## Vaginal infection and scoring for disease

Mice were injected subcutaneously with 2 mg of medroxyprogesterone and five days later inoculated intravaginally with 10 μl containing various concentrations of HSV-1 strain NS or HSV-2 strain MS [36, 37]. Mice were weighed daily for two weeks and virus cultures were obtained by swabbing the vagina at varying times post-infection. Some animals were sacrificed on day four to collect DRG and genital tract tissues. Animals were scored daily for genital disease on a scale of 0 to 4 by assigning 1 point each for erythema, perianal hair loss, urinary staining, and necrosis. Mice were humanely euthanized when they reached 20% body weight loss or a disease score of 3. DRG were collected and frozen at the end of the experiment.

## Virus cultures from vaginal swabs

Vaginal swabs were placed in one mL Dulbecco's modified Eagle's medium (DMEM) containing 5% fetal bovine serum supplemented with 25 μg/mL vancomycin [37]. For viral titers, 200 μl of undiluted and 10-fold serial dilutions of swab media were added to Vero cells for one hour at 37˚C, overlaid with 1.5% carboxy methylcellulose, and incubated for 72 hours. Plates were fixed and stained with 0.1% crystal violet. Plaques were counted and expressed as PFU/mL [37]. The limit of detection of the assay is 5 PFU/mL.

## HSV DNA isolation and real-time qPCR for HSV DNA copy number in DRG

Viral genomic DNA was isolated from mouse DRG samples and qPCR performed as previously described [26]. Taqman qPCR was used to amplify the HSV-1 Us9 gene (F: ACGGCC TCGCCAGTTTC, R: TTGGCCGCCTCGTCTTC, probe: 6FAM-TCGAAGCCTACTACT CG-MGBNFQ) or the HSV-2 Us9 gene (F: GGCAGAAGCCTACTACTCGGAAAA, R: CCATGCGCACGAGGAAA) from 5 μL of sample DNA. In a separate reaction, the mouse adipsin gene (F: GCAGTCGAAGGTGTGGTTACG, R: GGTATAGACGCCCGGCTTTT) was amplified from 5 μL of sample DNA. HSV DNA copy number was expressed as $\log_{10}$ HSV DNA copies per $10^5$ adipsin gene copies. Samples that did not yield a positive signal in duplicate wells by 40 cycles were considered negative.

## Endpoint dilution ELISA titers

Mice were bled 30 days after the final immunization. Microtiter plates were coated with 100 ng of purified gC1, gC2, gD1, gD2, gE1 or gE2 and IgG ELISA performed [25, 38]. Endpoint titers in immunized mice were calculated as the dilution that had at least a 2-fold higher OD value than Poly(C) control sera at the same dilution.

## Histopathology

Mice were sacrificed 4 days post-challenge, vaginal tissues removed and fixed for 48 hours at room temperature in freshly prepared 4% formaldehyde. The tissues were washed, dehydrated, cleared, and infiltrated with paraffin. The vaginal tissues were embedded to allow the entire

length of the vagina to be viewed in a single tissue block. Serial 4 μm sections were cut at two depths of the tissue block 100 μm apart. Slides were stained with hematoxylin and eosin and imaged on a Nikon Eclipse 1000 microscope. Images were obtained with 10, 20, and 40X objectives using a Spot model 4.2 CCD color camera and Spot version 4.04 software.

## Immunohistochemistry

Serial sections of vaginal tissue were deparaffinized and rehydrated. Sections were subjected to Heat Induced Epitope Retrieval in 10 mM Sodium Citrate buffer, 0.05% Tween-20, pH 6.0. The sections were heated and maintained at a temperature above 100˚ C for 10 mins, cooled, and permeabilized with 0.1% Triton-X-100 for 10 mins. The sections were washed and blocked with 10% normal goat serum. Rabbit anti-HSV-1 (Abcam) was used to detect HSV-1 antigen in the tissue. HSV-antibody negative, isotype-control IgG served as a control. Antibodies were incubated overnight at 4˚C, washed, and goat anti-rabbit biotin secondary antibody added for 60 min at room temperature followed by avidin-biotin-HRP complex (Thermo Fisher Scientific) for 30 min. 3,3′-Diaminobenzidine (Thermo Fisher Scientific) was used to detect the HSV-1 antigen, counter-strained with hematoxylin, dehydrated, cleared, and mounted with cytoseal 60 (Electron Microscopy Sciences). Slides were imaged with a 10X objective. To measure the number of HSV-1 antigen foci, slides were imaged with a 2X objective, imported into Image J and manually counted.

## Statistical analysis

The log-rank test was used to calculate $P$ values for survival curves. The Mann-Whitney test was used to calculate $P$ values in experiments evaluating vaginal titers, foci of disease by immunohistochemistry and virus DNA copy number in DRG. The two-tailed Student's t-test was used to calculate $P$ values comparing serum ELISA titers. All significance tests were two-tailed with an alpha value of 0.05. Analysis were done using GraphPad Prism version 6.0 (GraphPad software Inc).

## Author Contributions

**Conceptualization:** Kevin P. Egan, Sita Awasthi, Gary H. Cohen, Drew Weissman, Harvey M. Friedman.

**Formal analysis:** Kevin P. Egan, Harvey M. Friedman.

**Funding acquisition:** Gary H. Cohen, Drew Weissman, Harvey M. Friedman.

**Investigation:** Kevin P. Egan, Lauren M. Hook, Alexis Naughton, Sita Awasthi.

**Methodology:** Sita Awasthi, Harvey M. Friedman.

**Project administration:** Harvey M. Friedman.

**Resources:** Norbert Pardi, Gary H. Cohen, Drew Weissman, Harvey M. Friedman.

**Supervision:** Harvey M. Friedman.

**Visualization:** Kevin P. Egan, Harvey M. Friedman.

**Writing – original draft:** Kevin P. Egan, Harvey M. Friedman.

**Writing – review & editing:** Kevin P. Egan, Lauren M. Hook, Norbert Pardi, Sita Awasthi, Gary H. Cohen, Drew Weissman, Harvey M. Friedman.

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
