## [Decision Letter · Decision Letter 0]

25 Jun 2020

Dear Dr. Friedman,

Thank you very much for submitting your manuscript "An HSV-2 nucleoside-modified mRNA genital herpes vaccine containing glycoproteins C, D, and E protects mice against HSV-1 genital lesions and latent infection" for consideration at PLOS Pathogens. As with all papers reviewed by the journal, your manuscript was reviewed by members of the editorial board and by several independent reviewers. The reviewers appreciated the attention to an important topic. Based on the reviews, we are likely to accept this manuscript for publication, providing that you modify the manuscript according to the review recommendations.

Even so both reviewers felt that the mechanistic basis of the superiority of the RNA vaccination approach remains unclear, both value the reported data and just require some clarification of the methods.

Sincerely,

Christian Munz

Associate Editor

PLOS Pathogens

Erik Flemington

Section Editor

PLOS Pathogens

Kasturi Haldar

Editor-in-Chief

PLOS Pathogens

orcid.org/0000-0001-5065-158X

Michael Malim

Editor-in-Chief

PLOS Pathogens

orcid.org/0000-0002-7699-2064

Even so both reviewers felt that the mechanistic basis of the superiority of the RNA vaccination approach remains unclear, both value the reported data and just require some clarification of the methods.

Reviewer Comments (if any, and for reference):

Reviewer's Responses to Questions

**Part I - Summary**

Reviewer #1: The authors report a study regarding the use of two HSV-2 specific vaccination strategies, either RNA or protein based, using the HSV-2 glycoproteins C, D and E, as antigens. In addition to HSV-2 specific immune responses, also effects on HSV-1 infections were analysed in vivo.

This is a very interesting study and the authors provide evidence that a HSV-2 specific vaccination strategy protects from genital disease and furthermore, that also HSV-1 challenged animals are protected. In addition, they found the RNA vaccination approach to be even better, when compared with pure protein vaccination.

Reviewer #2: In this clearly written paper Egan et al describe the efficacy of a trivalent Herpes simplex viral vaccine containing glycoproteins C, D and E from HSV-2, expressed either as recombinant baculovirus proteins or as RNA, on genital HSV1 infection of mice and in comparison to effects on HSV2 infection. This extends previously published studies demonstrating greater protection of the RNA vaccine than the protein vaccine against HSV2 genital infection in mice, using similar study protocols. The studies are well conducted and results clear cut, although they do not necessarily explain previously published human trial results with HSV1 and 2 using a gD2-MPL vaccine. Both RNA and protein vaccines protected mice from HSV1 disease and death and infection of the DRG, although the RNA vaccine was more potent against vaginal virus titres two days after infection (but not at day 4). Somewhat surprisingly the RNA vaccine was less potent against HSV1 than HSV2 challenge in this parameter.

The paper is short and it seems there would more opportunity to study mechanisms than is provided.

Specific Questions:

In Figure 2D how were the foci counted. Were multiple sections taken at different parts of the vagina? Was manual or computer aided counting per section done? Was there any variability throughout the vagina? Was there any T cell or other immune cell infiltrate in the vaccine treated mice?

In figure 3C: vaginal HSV1 titres at day 2 are on average >1 log higher than in Figure 1E despite using the same inoculum size. Is there a reason for this variability?

Figure 4B: Are they surprised there is no inoculum dose effect on vaginal viral titres?

Discussion: Are they surprised that the vaginal HSV1 replication out to day 4 in Figure 1E does not result in DRG infection? Do they have an explanation? Presumably this is all local vaginal replication without any axonal-DRG circuit?

Some omissions:

Given the results in Figures 3C and 4C why not use gD2 alone as an immunogen in an extra arm to see if it mimics the Herpevac results regarding protection against HSV1 versus HSV2?

There is no evaluation of T cell responses, even follicular T helper cell responses, in blood or tissue. This might be expected, given the use of adjuvants.

**Part II – Major Issues: Key Experiments Required for Acceptance**

Reviewer #1: The authors found the RNA vaccination approach to be even better, when compared with pure protein vaccination.

The reason for this are not completely clear. Could it be that the RNA backbone induces a stronger immune response via a TLR activation of antigen presenting cells? The authors should look at this in more detail.

In addition and very interestingly the authors report, that this HSV-2 specific vaccination strategy also generates cross-reactive antibodies, neutralizing HSV-1.

It would be interesting to analyse if these neutralizing antibodies protect HSV-1/2 infected animals also in a therapeutic setting, not only in a prophylactic vaccination strategy.

Reviewer #2: Given the results in Figures 3C and 4C why not use gD2 alone as an immunogen in an extra arm to see if it mimics the Herpevac results regarding protection against HSV1 versus HSV2?

**Part III – Minor Issues: Editorial and Data Presentation Modifications**

Reviewer #1: Minor: lane 260 (Fig 4F should be Fig 4C)

Reviewer #2: Minor: Does poly C induce interferon?

PLOS authors have the option to publish the peer review history of their article (what does this mean?). If published, this will include your full peer review and any attached files.

Reviewer #1: **Yes: **Alexander Steinkasserer

Reviewer #2: No
---

## [Editor Report · Decision Letter 1]

9 Jul 2020

Dear Dr. Friedman,

We are pleased to inform you that your manuscript 'An HSV-2 nucleoside-modified mRNA genital herpes vaccine containing glycoproteins gC, gD, and gE protects mice against HSV-1 genital lesions and latent infection' has been provisionally accepted for publication in PLOS Pathogens.

Best regards,

Christian Munz

Associate Editor

PLOS Pathogens

Erik Flemington

Section Editor

PLOS Pathogens

Kasturi Haldar

Editor-in-Chief

PLOS Pathogens

orcid.org/0000-0001-5065-158X

Michael Malim

Editor-in-Chief

PLOS Pathogens

orcid.org/0000-0002-7699-2064
---

## [Editor Report · Acceptance letter]

20 Jul 2020

Dear Dr. Friedman,

We are delighted to inform you that your manuscript, "An HSV-2 nucleoside-modified mRNA genital herpes vaccine containing glycoproteins gC, gD, and gE protects mice against HSV-1 genital lesions and latent infection," has been formally accepted for publication in PLOS Pathogens.

Best regards,

Kasturi Haldar

Editor-in-Chief

PLOS Pathogens

orcid.org/0000-0001-5065-158X

Michael Malim

Editor-in-Chief

PLOS Pathogens

orcid.org/0000-0002-7699-2064